# Hydrolysis Mechanism of Carbamate Methomyl by a Novel Esterase PestE: A QM/MM Approach

**DOI:** 10.3390/ijms24010433

**Published:** 2022-12-27

**Authors:** Zijian Wang, Qingzhu Zhang, Guoqiang Wang, Wenxing Wang, Qiao Wang

**Affiliations:** Big Data Research Center for Ecology and Environment, Environment Research Institute, Shandong University, Qingdao 266003, China

**Keywords:** methomyl, hydrolysis, esterase, biotransformation, nucleophilic attack, quantum mechanics/molecular mechanics, molecular dynamics

## Abstract

Methomyl is one of the most important carbamates that has caused potential hazardous effects on both human beings and the environment. Here, we systematically investigated the hydrolysis mechanism of methomyl catalyzed by esterase PestE using molecular dynamics simulations (MD) and quantum mechanics/molecular mechanics (QM/MM) calculations. The hydrolysis mechanism involves two elementary steps: (Ⅰ) serine-initiated nucleophilic attack and (Ⅱ) C-O bond cleavage. Our work elicits the atomic level details of the hydrolysis mechanism and free energy profiles along the reaction pathway. The Boltzmann-weighted average potential barriers are 19.1 kcal/mol and 7.5 kcal/mol for steps Ⅰ and Ⅱ, respectively. We identified serine-initiated nucleophilic attack as the rate determining-step. The deep learning-based *k*_cat_ prediction model indicated that the barrier of the rate-determining step is 15.4 kcal/mol, which is in good agreement with the calculated results using Boltzmann-weighted average method. We have elucidated the importance of the protein–substrate interactions and the roles of the key active site residues during the hydrolysis process through noncovalent interactions analysis and electrostatic potential (ESP) analysis. The results provide practical value for achieving efficient degradation of carbamates by hydrolases.

## 1. Introduction

Methomyl is a carbamate first synthesized by Du Pont Company in 1966, and has been applied to control invading organisms in agriculture around the world [1,2,3]. Methomyl is still extensively produced and employed nowadays, although it was forbidden in many countries [4]. According to Sapec Agro Portugal, the active ingredient concentration of methomyl can reach levels higher than 830 mg/L per hectare currently [5]. Because of high solubility (57.9 g/L, 20 °C) and low soil affinity, methomyl can easily cause contamination of groundwater and surface water in the natural environment [6]. Continuous exposure to methomyl further results in accumulation in human tissue, including blood, kidney, liver and brain [3,7,8,9]. Methomyl can cause various health hazards, such as hepatotoxicity, neurotoxicity, and cytotoxicity [10]. Therefore, it is important to investigate the environmental behavior of methomyl and further assess its persistence, ecological risk and adverse effects.

Catalytic reactions are widely used in the transformation of various organic maters [11,12,13,14]. Among catalytic reactions, hydrolysis is one of the most efficient catalytic reactions. Hydrolysis is considered to be a crucial degradation pathway for methomyl in the natural environment because it has a carboxylic ester bond in the molecular structure. The hydrolysis products (methomyl oxime) were observed to be the largest pesticide concentration (40 μg/L) among 90 pesticide transformation products in a nationally distributed synoptic assessment of contaminant exposures [15]. Therefore, the investigation of hydrolysis of methomyl is important and necessary for assessing their environmental persistence and hazard.

Esterase is a class of hydrolase family with the ability to hydrolyze ester group that is successfully applied to degrade environmental pollutants [16,17]. Xu and co-workers reported a strain named MDW-1, isolated from the enrichment of activated sludge, could transform methomyl to S-methyl-N-hydroxythioacetamidate [18]. Zhang and co-workers showed that two bacterial strains named MDW-2 and MDW-3, isolated from the enrichment, can convert methomyl to methomyl oxime and methylcarbamic acid, which can later be used as the carbon source for the growth of MDW-2 and MDW-3 [19]. Unfortunately, the crystal structures of these strains have not been resolved so far. However, Fan and co-workers reported that an extremely thermophilic esterase PestE, isolated from Pyrobaculum calidifontis VA1, could be used as the template for hydrolysis of the organic pesticides [20,21]. The core domain of PestE structure is a classical α/β-hydrolase fold structure with a cap structure at the C-terminal of the β-sheet. The catalytic active center of esterase PestE consists of the catalytic triad (Ser157, His284, and Asp254) and the oxyanion hole (Gly85 and Gly86) [21]. The structural diagram of secondary structure of esterase PestE and active amino acid residues was shown in Figure 1. Although the hydrolysis of methomyl has been experimentally studied in previous studies, the detailed binding interaction and roles of key amino acids during catalysis are still unknown. Due to the high rate of the enzyme–catalyzed reaction, the intermediates in the reaction process were rapidly decomposed and cannot be found in the environment, and these intermediates are indispensable for understanding the reaction mechanism.

Quantum mechanics/molecular mechanics (QM/MM) method is a computational method that combines the accuracy of quantum mechanics with the speed advantage of molecular mechanics. In the QM/MM approach, a small part of the system is modeled quantum-mechanically, while the protein and solvent are treated by molecular mechanics [22]. QM/MM method can provide atomic-level details of enzyme catalysis mechanisms, and it is increasingly applied in computational enzyme chemistry. In the present study, the hydrolysis mechanism of esterase PestE towards methomyl was explored by using the molecular dynamics (MD) simulations and quantum mechanics/molecular mechanics (QM/MM) method. Two elementary steps are investigated in our simulation: (Ⅰ) serine-initiated nucleophilic attack and (Ⅱ) C-O bond cleavage. In addition, the reduced density gradient (RDC) function and electrostatic potential (ESP) analysis were performed to investigate the protein–substrate interactions and the roles of the key active site residues during hydrolysis process. 

## 2. Results and Discussion

### 2.1. Molecular Dynamics 

MD simulations were conducted to investigate the configuration changes of esterase PestE and the substrate. The stability of the system can be evaluated by the root-mean-square deviation (RMSD) and cluster analysis with TTclust [23]. As shown in Appendix A, the RMSD among conformations maintains within 0.16 Å during the dynamic process, illustrating that the system is well-consistent. Same results can be obtained in Appendix A. Appendix A shows the dendrogram of the system, which can be used to choose the appropriate clusterization level and describe the relationship among conformations in each cluster. Appendix A presents the ratio of different clusters during the dynamic process. The ratio of the fifth cluster is the largest, which indicates that it is relatively stable. Appendix A presents the 2D projection plot of the relative distances among clusters. The maximum and minimum relative distances among clusters are 0.77 and 0.74, respectively. The maximum and minimum RMSD values among clusters are 1.27 Å and 0.78 Å, respectively. According to the results of cluster analysis, dynamic trajectories can be classified thoroughly, which enables us to select reliable conformations for QM/MM simulations. 

### 2.2. Reaction Mechanism and Energy Profile

As shown in Figure 2a, the hydrolysis mechanism of esterase PestE towards methomyl involves two concerted steps. First, the proton (H^1^) of Ser157 transfers to His284 while the oxygen atom (O^1^) acts as a nucleophile to attack the carbonyl carbon (C^1^) of the substrate. Subsequently, the C^1^-O^3^ bond of the substrate breaks while the lone pair of the oxygen atom (O^3^) attracts the hydrogen atom (H^1^) from His284. We selected five conformations from the MD trajectories for QM/MM calculations. The rate constant of an enzyme-catalyzed reaction generally exhibits a wide range of fluctuations. It is hypothesized that the snapshots obtained by MD simulations correspond to the local rate constant [24,25]. Hence, the Boltzmann-weighted average method was used to calculate the average energy barrier, which can be described according to the following equation:
$$\Delta E=-RT\ln\left\{\frac{1}{n}\overset{n}{\underset{i=1}{\sum }}\exp\left(\frac{-\Delta E_{i}}{RT}\right)\right\}$$
where Δ*E* is the average energy barrier, *R* is the gas constant, *T* is the temperature, *n* is the number of snapshots, Δ*E_i_* is the energy barrier of path *i*.

The reaction energy barriers of five snapshots calculated from QM/MM were plotted in Figure 2b. We calculated the Boltzmann-weighted average barriers of five snapshots to avoid mistakes caused by certain factors [16]. In the first step, the H^1^ of Ser157 transfers to His284, while the O^1^ on the Ser157 attacks the substrate. The Boltzmann-weighted average barrier of this step is 19.1 kcal/mol, thus it is the rate-determining step of the reaction. We have also carried out the deep learning-based *k*_cat_ prediction model [15] to predict energy barrier. The results indicated that the barrier of the rate-determining step of the reaction is 15.4 kcal/mol, which is in good agreement with the calculated results using Boltzmann-weighted average method. In the second step, the C^1^-O^3^ bond of the acylase intermediate cleaves, while the proton H^1^ from His284 transfers to O^3^ of the substrate of which the Boltzmann-weighted average barrier is 7.5 kcal/mol. 

### 2.3. Reaction Structure Details

In order to figure out the details of the catalytic reaction process, we focused on the analysis of changes in structures (bond length, bond angle, dihedral angle) of the intermediates and transition states. As shown in Figure 3, the key structure change details of the snapshot along the lowest energy barrier pathway were provided during the enzyme-catalytic reaction process. The key structure change details of other snapshots are described in Appendix A. In the first step, we optimized the reactant and found that the distance between Ser157 and substrate (C^1^-O^1^) is 2.57 Å. Subsequently, after a transition state, the distance of C^1^-O^1^ decreases to 1.51 Å, while the angle (O^1^-C^1^-O^3^) changes from 84.77° to 103.39° and the angle (O^1^-C^1^-O^2^) changes from 99.31° to 111.34°, suggesting that the enzyme-substrate complex has formed. Meanwhile, Ser157 is deprotonated by His284 and the distance between H^1^ and N^1^ changes from 1.85 Å to 1.07 Å. In the second step, the distance between C^1^ and O^3^ increases from 1.43 Å to 2.67 Å, indicating that the C^1^-O^3^ bond of the substrate breaks. In addition, the angle (O^1^-C^1^-O^3^) of the substrate changes from 103.39° to 76.11°, and the dihedral angle (O^1^-C^1^-O^3^-H^1^) increases from 26.68° to 37.27°. All changes above are consistent with the changes in distance. While the C^1^-O^3^ bond of the substrate is breaking, the length of H^1^-O^3^ decreases from 3.25 Å to 1.00 Å, which indicates a hydroxylated product is forming. During the whole process, the hydrogen bond among the key amino acid residues Ser157, His284 and Gly85 around the substrate has the ability to stabilize the substrate and promote the catalytic reaction. Specifically, the N-H group of Gly85 and the O^2^ atom of substrate maintain at approximately 2 Å, invariably performing as a strong hydrogen bond.

### 2.4. Noncovalent Interactions

Noncovalent interactions dominate chemical interactions between a protein and a drug, or a catalyst and its substrate, and even several chemical reactions [26,27]. Johnson and co-workers designed a visual method for studying noncovalent interactions [26], which is based on the electron density and its derivatives in real space. Reduced density gradient (RDG) is a dimensionless quantity derived from electron density *ρ(r)* and its first derivative,
$$RDG\left(r\right)=\frac{1}{2{\left(3\pi^{2}\right)}^{1/3}}\frac{\left|\nabla \rho \left(r\right)\right|}{\rho {\left(r\right)}^{4/3}}$$

The nature and strength of noncovalent interactions shown in the molecules can be comprehended by mapping *ρ(r)* against sign (λ_2_) [28]. The sign of λ_2_ is used to distinguish repulsive (λ_2_ > 0, unbonded) and attractive (λ_2_ < 0, bonded) interactions. The *ρ(r)*sign(λ_2_) peaks reveal the strength of noncovalent interactions. With the help of Multiwfn 3.7 [29] and VMD 1.9.3 [30], the color-filled RDG iso-surfaces and the scatter plots of weak interactions between methomyl and key amino acid residues were drawn for the structure of R, TS1 and TS2.

Figure 4(a1,b1,c1) show the scatter plots of weak interactions between the substrate and three key residues. The abscissa and ordinate represent the value of *ρ(r)*sign(λ_2_) and RDG function, respectively. The interaction regions are labeled blue, green, and red for attractive, van der Waals (vdW), and repulsive interactions, respectively. Figure 4(a2,b2,c2) display the color-filled RDG iso-surfaces, and blue regions represent for attractive interactions, the green regions for Vander Waals interactions and the red regions for repulsive interactions. As shown in Figure 4(a1), the peak value ranges from −0.04 to −0.03 a.u., indicating that the attraction between O1 and C1, which promotes the occurrence of the nucleophilic attack reaction. Comparing Figure 4(b1) with Figure 4(b2), we can find that the interactions of proton (H^1^) transfer and nucleophilic attack both exhibit a large red-blue annular region, illustrating that they happen at the same time. In addition, the steric effect between Ser157 and His284 is significantly enhanced, we conclude that the system is developing to a steady state while the oxygen (O^1^) of Ser157 combines with the substrate. It is worth mentioning that the phenomenon concluded from Figure 4(b1,b2) can also correspond to Figure 4(a1,a2). In addition, comparing Figure 4(b2) with Figure 4(c2), we can conclude that the interactions between C^1^ and O^3^ become stronger, displaying a larger red-blue annular region, which indicates that the bond between C^1^ and O^3^ will break. Meanwhile, H^1^ and O^3^ appear a large red-blue annular region, which implies that the second reaction is also a concerted reaction. Noted that the hydrogen bond interaction between substrate and Gly85 exists invariably, which stabilizes the substrate and contributes to the catalytic reaction process. It can be seen that the noncovalent interaction of attraction and repulsion play a central part in the catalytic reaction.

### 2.5. Electrostatic Potential

Electrostatic potential (ESP) is a concept in wave function analysis, which is of great significance to investigate electrostatic interactions between molecules and to predict reaction sites and molecular properties [29,31,32]. For the propose of illustrating the electrostatic interactions among molecules and figuring out reactive sites, the Electrostatic potential (ESP) was calculated by Mutiwfn 3.7 [29] and figured by VMD 1.9.3 [30]. 

The ESP distribution on the van der Waals (vdW) surface and extreme points of TS1 is exhibited in Figure 5a, noting that the blue and golden spheres, respectively, refer to the surface local minimum and maximum points of ESP. In addition, the positive extreme points represent that the electrostatic potential in this area is dominated by the nuclear charge, and the negative extreme points represent that the contribution of electrons is greater. According to Figure 5a, the significant ESP minimum points on the surface contain the hydroxyl hydrogen, methyl group and hydrogen bonds, and the significant ESP maximum points are attributed to the hydroxyl oxygen and nitrogen lone pair electrons. Interestingly, the ESP absolute value surrounding O^3^ (−60.9 kcal/mol) is large, which provides a new idea that O^3^ will undergo a proton transferring process, and we also verified this idea through QM/MM calculations. The vdW surface areas in different ESP ranges are shown in Figure 5b. According to Figure 5b, we can find that the values of ESP between −15.7 and 9.7 kcal/mol account for 51.1% of the surface areas, and the positive and negative ESP take up approximately 45% and 55% of the total areas, respectively. 

## 3. Computational Methods

### 3.1. System Preparation and MD Simulations

The initial structure studied was selected from the crystal structure of esterase PestE (PDB code: 3ZWQ, resolution: 2.0 Å, http://www.rcsb.org/ (accessed on 17 October 2020)) [21]. Methomyl was docked into the catalytic active center of esterase PestE via Autodock Vina and AutoDock Tools [33,34]. In the docking simulation, the grid box size was set 25 × 25 × 25 Å centered around the Ser 157 of esterase PestE. Nine docking models were obtained, and the best binding conformation was selected for the subsequent MD simulations. The topology and parameter of methomyl was generated by Swissparam [35]. The protonation states of the amino acid residues were considered by PROKA program on account of the pKa values [36]. The enzyme–substrate complex structure was optimized by the HBULD facility of CHARMM package [37,38]. The structure was dissolved in a TIP3P water sphere model with a diameter of 65 Å [39]. In order to maintain electrical neutrality, two sodium ions were stochastically added to the structure. The structure contains 14123 atoms. Firstly, the structure was heated from 0 K to 298.15 K in 50 ps, then it was equilibrated in 500 ps with a 1 fs time step, followed by a 30 ns stochastic boundary molecular dynamic (SBMD) simulation using canonical ensemble NVT at 298.15 K [2,37,40,41,42]. In the SBMD simulations, Langevin dynamics and Leap-frog algorithm of CHARMM package were implemented. Cluster analysis on 3000 conformations obtained by MD simulations was performed via TTClust [23], and the conformations were selected from the stable clusters for QM/MM calculations.

### 3.2. QM/MM Methodology

The QM/MM calculations were performed by the ChemShell 3.6.0 platform [43] that combines the quantum chemistry program Turbomole 7.2 [44] with the molecular mechanics module DL-POLY [45]. The electrostatic field embedding and charge shift model were used during QM/MM calculations [46,47]. The QM region (61 atoms) contains the substrate methomyl, Gly85, Ser157, His284. The geometry optimizations of intermediates and transition states were performed at the M06-2X/6-31G(d,p) level, and single-point energies were calculated at the M06-2X/6-311G(d,p) level. The CHARMM27 force field was used for the MM region. During the QM/MM calculation, the atoms within 20 Å of the substrate were allowed to move. 

## 4. Conclusions

Using MD and QM/MM approaches, we have elucidated the hydrolysis mechanism and the structural details of esterase PestE reacting with environmental pollutant methomyl. The hydrolysis process involves two steps. In the first step, the H^1^ of Ser157 transfers to His284, while the O^1^ on the Ser157 attacks the substrate. The Boltzmann-weighted average barrier of this step is 19.1 kcal/mol, and it is the rate-determining step of the reaction. In the second step, the C^1^-O^3^ bond of the acylase intermediate cleaves while the proton H^1^ from His284 transfers to the O^3^ of the substrate, the Boltzmann-weighted average barrier of this step is 7.5 kcal/mol. Through noncovalent interactions analysis and electrostatic potential (ESP) analysis, we have confirmed the importance of the protein–substrate interactions and the roles of key active site residues during the hydrolysis process. The active site residues His284 and Ser157 participate in the proton transfer and nucleophilic during the catalysis process. Another active residue Gly85 stabilizes the substrate and enhances the enzymatic efficiency. We believe that these results are useful to interpret crystal structures and kinetic experiments of methomyl hydrolysis. Further, the atomic level details obtained in this study can facilitate designing a rational enzyme engineering strategy to efficiently degrade pesticides, thus generating benefits for environmental sustainability. 

## Figures and Tables

**Figure 1 ijms-24-00433-f001:**
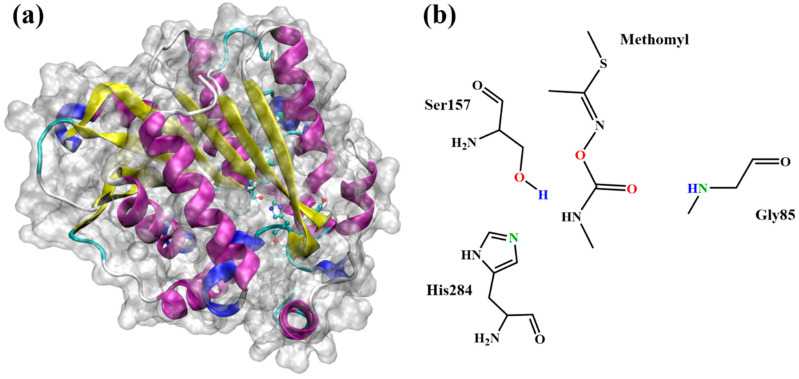
(**a**) Structural diagram of secondary structure of esterase PestE. (**b**) The chemical structure of methomyl and a representation of QM region.

**Figure 2 ijms-24-00433-f002:**
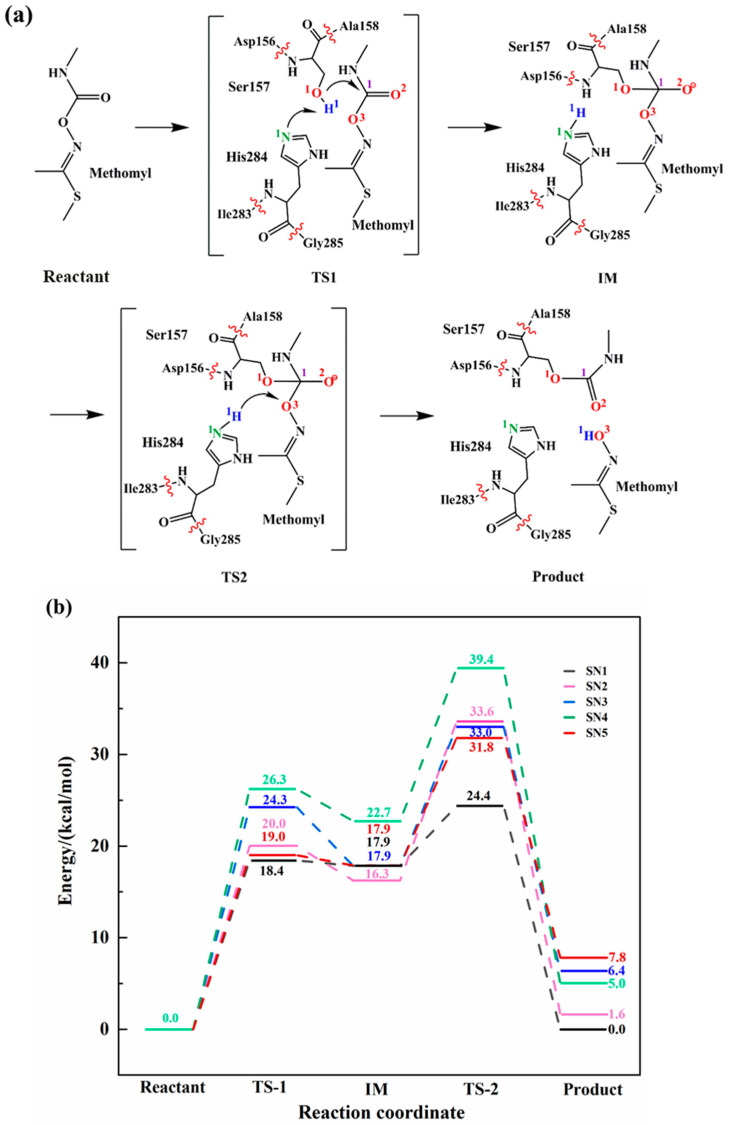
Reaction mechanism and energy profile of enzymatic catalysis reaction. (**a**) The proposed mechanism of the biodegradation of methomyl; (**b**) Energy profiles of 5 snapshots with different color.

**Figure 3 ijms-24-00433-f003:**
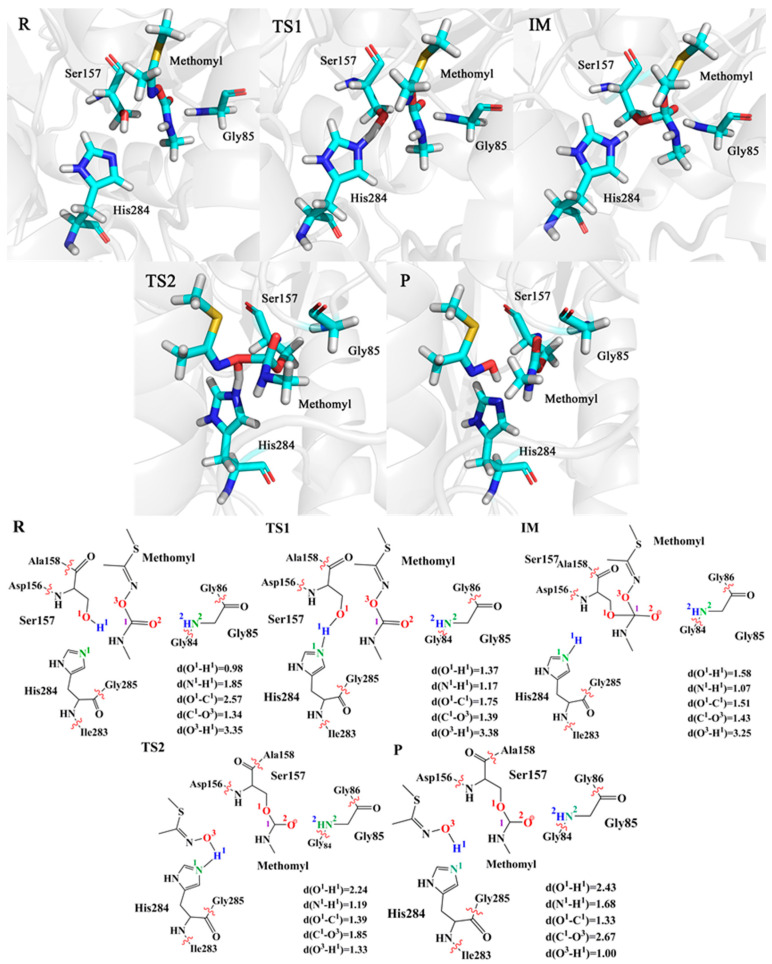
Optimized structures and key bond distances for the R, TS1, IM, TS2 and P. The unit of distance is Å.

**Figure 4 ijms-24-00433-f004:**
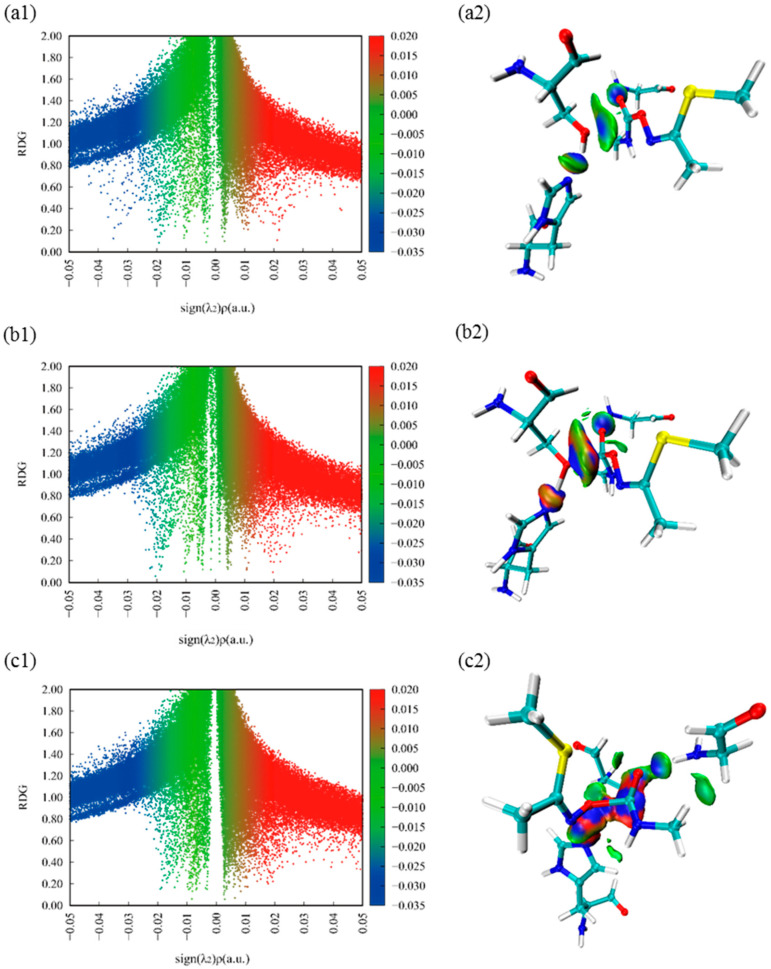
Color-mapped reduced density gradient (RDG) iso-surface graphs and scatter plots of methomyl and three critical amino acids. Species shown: (**a1**), (**b1**), (**c1**) show the color-mapped reduced density gradient (RDG) scatter plots of methomyl and three critical amino acids of R, TS1, TS2, respectively. (**a2**), (**b2**), (**c2**) show the color-mapped reduced density gradient (RDG) iso-surface graphs of methomyl and three critical amino acids of R, TS1, TS2, respectively.

**Figure 5 ijms-24-00433-f005:**
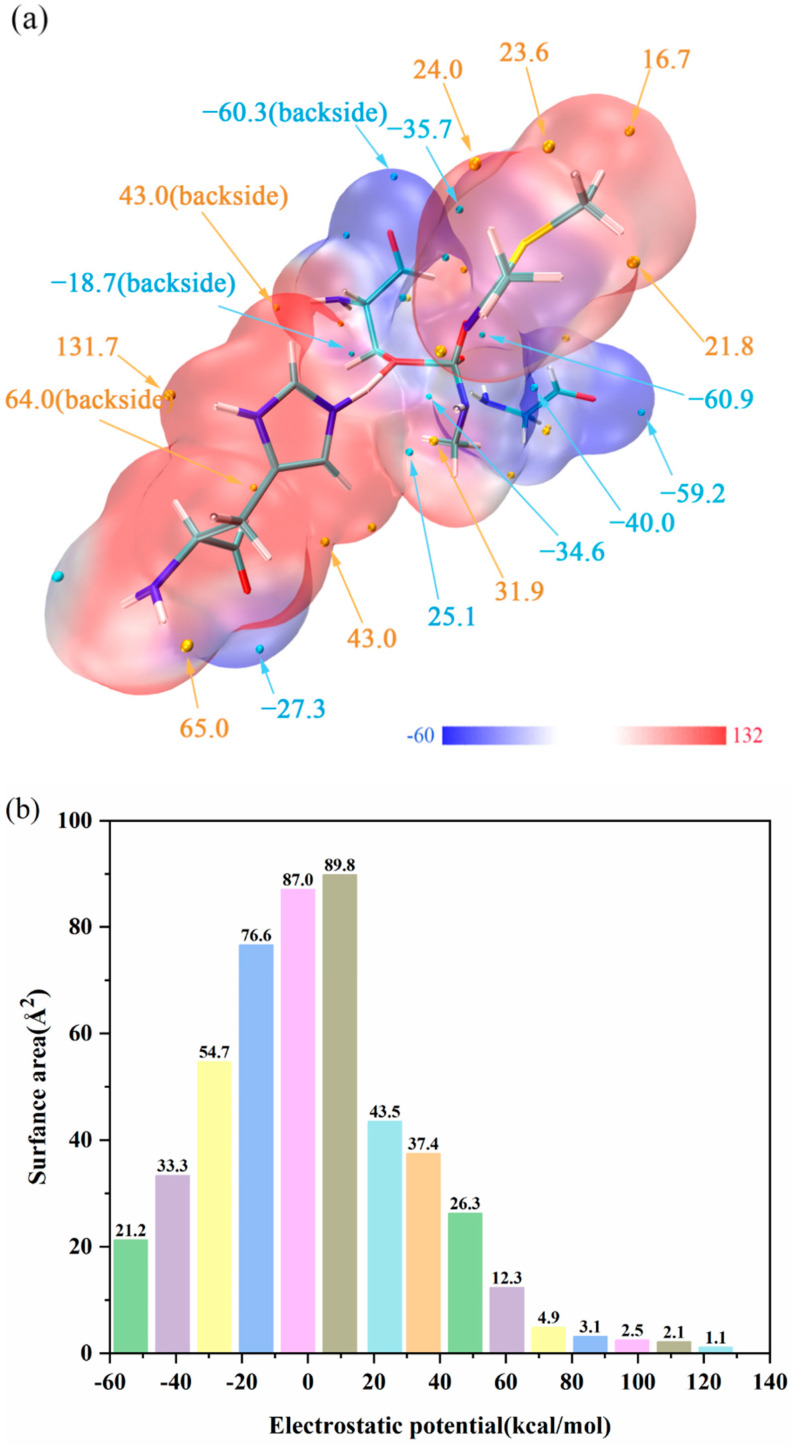
(**a**) ESP distribution on molecular vdW surface of TS1. The unit is kcal/mol, and important surface local minima and maxima of ESP are represented as blue and golden spheres, respectively. (**b**) Surface area in each ESP range on the vdW surface of the system.

## Data Availability

Not applicable.

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
