# Peer review of "Hydrolysis Mechanism of Carbamate Methomyl by a Novel Esterase PestE: A QM/MM Approach"

_ijms, 2022, doi:10.3390/ijms24010433_

Round 1

Reviewer 1 Report

Herein, the author has investigated the 9 hydrolysis mechanism of methomyl catalyzed by esterase PestE using molecular dynamics simulations and quantum mechanics/molecular mechanics (QM/MM) calculations. The hydrolysis mechanism involves two elementary steps: (â… ) serine-initiated nucleophilic attack and (â…¡) C-O bond cleavage. Our work elicits the atomic level details of the hydrolysis mechanism and free energy profiles along the reaction pathway. The Boltzmann-weighted average potential barriers were 19.1 kcal/mol and 7.5 kcal/mol for steps â…  and â…¡, respectively. Still, there is room for improvement, thus, I would suggest a minor revision of the manuscript to improve the quality of the article for its possible publication in this reputed journal. Kindly find my comments below:

1.     Keywords should be updated with more relevant terms.

2.     The English of the manuscript needs to be improved.

3.     Several typo errors and spelling mistakes in the manuscript such as importort which should be important. Kindly revise the whole manuscript and correct such errors.

4.     Discuss more about hydrolysis with the support of recent literature.

5.     Although this paper focus on Quantum mechanics/molecular mechanics (QM/MM) but there is minimal discussion of this method in the introduction. Hence, author must elaborate more about this in the introduction section.

6.     Figure 1 is blurry; therefore, a more quality image must be added with an elaborate discussion of the findings attained through it. Same for Figure 2 and 3.

7.     Abbreviations must be mentioned earlier in their first place.

8.     Why in 2.1. the structure’s system was heated upto 298.15K? Discuss.

9.     Was the structure included in the system static or dynamic?

10.  A lot of spelling errors in figures such as rectant in figure 2. Kindly correct such errors in whole manuscript.

11.  Discuss more about noncovalent interactions. How these interactions plays crucial role in this specific study?

12.  References need to be revised and check. The author is suggested to add literature in the article: Coatings 12, no. 10 (2022): 1459, https://doi.org/10.3390/coatings12101459; Process Safety and Environmental Protection 161 (2022): 801-818, https://doi.org/10.1016/j.psep.2022.03.082, Sustainable Chemistry and Pharmacy 29 (2022): 100785, https://doi.org/10.1016/j.scp.2022.100785, Journal of Bio-and Tribo-Corrosion 7, no. 2 (2021): 1-48, https://doi.org/10.1007/s40735-021-00501-y; Current Nanoscience 18, no. 2 (2022): 203-216, https://doi.org/10.2174/1573413717666210216120741; Elsevier, 2020. 345-362, https://doi.org/10.1016/B978-0-12-819359-4.00018-0;  Surfaces and Interfaces 20 (2020): 100542, https://doi.org/10.1016/j.surfin.2020.100542, Arabian Journal for Science and Engineering 45, no. 6 (2020): 4773-4783, https://doi.org/10.1007/s13369-020-04514-6, Materials Research Express 7, no. 1 (2019): 016510, https://doi.org/10.1088/2053-1591/ab5c6a., https://doi.org/10.1016/B978-0-12-819359-4.00018-0.

Author Response

Response to Reviewer 1 Comments:

Point 1: Keywords should be updated with more relevant terms.

Response 1: Thank you for the valuable comment. The keyword nucleophilic attack, biodegradation and molecular dynamics were added. The new keywords can be described as: methomyl; hydrolysis; esterase; biodegradation; nucleophilic attack; quantum mechanics/molecular mechanics; molecular dynamics.

Point 2: The English of the manuscript needs to be improved.

Response 2: Thank you for your reminder. We regret there were problems with the English. The manuscript has been carefully revised by a native English speaker to improve the grammar and readability. We believe that the language is now acceptable for the review process.

Point 3: Several typo errors and spelling mistakes in the manuscript such as importort which should be important. Kindly revise the whole manuscript and correct such errors.

Response 3: Thank you for your careful checks. We are sorry for our carelessness. Based on your comment, we have revised “importort” to “important” (line 38). In addition, we have carefully checked the manuscript and corrected the typo errors and spelling mistakes accordingly. In this revised version, corrections to our manuscript were all marked.

Point 4: Discuss more about hydrolysis with the support of recent literature.

Response 4: Thank you for your valuable comment. We have added discussion about hydrolysis. This change is in line 40-42.

Point 5: Although this paper focus on Quantum mechanics/molecular mechanics (QM/MM) but there is minimal discussion of this method in the introduction. Hence, author must elaborate more about this in the introduction section.

Response 5: Thank you for your nice suggestions on our paper. Accordingly, we have elaborated more about the Quantum mechanics/molecular mechanics (QM/MM). Quantum mechanics/molecular mechanics (QM/MM) method is a computational method that combines the accuracy of quantum mechanics with the speed advantage of molecular mechanics. In the QM/MM approach, a small part of the system is modeled quantum-mechanically, while the protein and solvent are treated by molecular mechanics(Dos Santos et al., 2022). The relevant descriptions was added in the revised manuscript (line 70-74).

Point 6: Figure 1 is blurry; therefore, a more quality image must be added with an elaborate discussion of the findings attained through it. Same for Figure 2 and 3.

Response 6: Thank you for pointing this out. We have uploaded more quality image for all figures in the revised manuscript. In addition, we have moved Figure 1. “TTcluster was applied to RMSD and cluster analysis of 30 ns MD simulations performed with CHARMM” to Supporting Information. Insteadlly, “(a) Structural diagram of secondary structure of esterase PestE and (b) The chemical structure of methomyl and a respresentation of QM region” was named as Figure 1. This change is in line 135.

Point 7: Abbreviations must be mentioned earlier in their first place.

Response 7: Thanks for your careful checks. We double-checked the abbreviations in this manuscript to ensure the right use of the abbreviations. For example, the abbreviations “TPs” in line 46 has been changed to “transformation products”. Moreover, abbreviations “RDG” in line 197 have been changed to “Reduced density gradient (RDG)”. Several similar problems have been corrected.

Point 8: Why in 2.1. the structure’s system was heated up to 298.15K? Discuss.

Response 8: Thank you for the valuable comment. In natural environment, the biotransformation process of methomyl occurs at normothermia. Accordingly, the structure’s system was heated up to 298.15K (25℃). In addition, we have cited relevant literatures (Bashir et al., 2020a; Bashir et al., 2020b; Brooks et al., 1983; Omeiri et al., 2022; Parveen et al., 2019). This change is in line 101.

Point 9: Was the structure included in the system static or dynamic?

Response 9: Thank you for this valuable comment. The structure included in the system is dynamic. Firstly, molecular dynamics was performed by CHARMM, which is a general method that encompasses any infinite or finite periodic tiling of a simulation. Secondly, representative conformations were selected for subsequent QM/MM calculations.

Point 10: A lot of spelling errors in figures such as rectant in figure 2. Kindly correct such errors in whole manuscript.

Response 10: Thank you for your correction. According to your suggestion, we have corrected “rectant” in figure 2 to “reactant”. Such errors have corrected after multiple checks. We believe that the spelling and language is now acceptable for the review process.

Point 11: Discuss more about noncovalent interactions. How these interactions plays crucial role in this specific study?

Response 11: Thank you again for your valuable suggestion. We have added discussion about noncovalent interactions in line 209-211 and line 223-224. We think that the modification of this discussion about noncovalent interactions is now acceptable for the review process.

Point 12: References need to be revised and check. The author is suggested to add literature in the article: Coatings 12, no. 10 (2022): 1459, https://doi.org/10.3390/coatings12101459; Process Safety and Environmental Protection 161 (2022): 801-818, https://doi.org/10.1016/j.psep.2022.03.082, Sustainable Chemistry and Pharmacy 29 (2022): 100785, https://doi.org/10.1016/j.scp.2022.100785, Journal of Bio-and Tribo-Corrosion 7, no. 2 (2021): 1-48, https://doi.org/10.1007/s40735-021-00501-y; Current Nanoscience 18, no. 2 (2022): 203-216, https://doi.org/10.2174/1573413717666210216120741; Elsevier, 2020. 345-362, https://doi.org/10.1016/B978-0-12-819359-4.00018-0;  Surfaces and Interfaces 20 (2020): 100542, https://doi.org/10.1016/j.surfin.2020.100542, Arabian Journal for Science and Engineering 45, no. 6 (2020): 4773-4783, https://doi.org/10.1007/s13369-020-04514-6, Materials Research Express 7, no. 1 (2019): 016510, https://doi.org/10.1088/2053-1591/ab5c6a., https://doi.org/10.1016/B978-0-12-819359-4.00018-0.

Response 12: Thank you for your reminder. We have doubled-revised and check all references. In addition, we have added all literatures in the article as suggested.

Reviewer 2 Report

The manuscript "Hydrolysis mechanism of carbamate methomyl by a novel esterase PestE: A QM/MM approach" studies in detail a common degradation pathway of a toxic compound, carbamate methomyl catalized by PestE, a novel - experimentally found - esterase. 

I think it is a novel study with very nice results, however the conclusions have been extracted based on a limited set of structures and the analysis methodology could be better explained to show its relevance. It would strengthen the points made if the authors could use more structures per cluster to show the same results. I am also missing a benchmark of several functionals or at least some solid justification to use the combination M06-2X/6-31G** (or the triple Z equivalent) for the QM/MM part.

Other comments:

1. Figure 1 should have the structure of the compounds studied and maybe the QM/MM zones, it would be clearer to the reader if they didn't have to scroll to page 5 to the mechanism picture.

2. In the computational details, the docking parameters used for Autodock should be specified, as well as the thermostat type for the MD simulations. In general, more details wouldnt hurt in this section. 

3. Figure 1 could be moved to the Supp Info or its results compressed to one single picture. 

4. line 119, "more scientifically" should be changed by "thoroughly"

5. I do not understand why the authors benchmark their energy error against a machine learning method. Which error does the ML method have? could the authors show error bars in their own results?

6. Line 159 in "spacial structures" , spacial is redundant and should be removed

7. Line 184, in "some chemical reactions", "some" sounds colloquial. The authors may consider rewriting that sentence.

8. Line 187, the acronym RDG needs to be explained and after "electron density" the greek letter rho could be added to refer to the next equation.

9. Line 191, what is sign (lambda) ? In general the last three points show that the part of "Covalent Interactions" could be improved to make it more understandable to the readers.

10. Line 209 refers to an unexistent Fig.4(b3)

11. Line 218: specied -> species

12. In Figure 4, the z axis seems to give the same information as the x axis. If this is not the case a clarification is needed

13. Figures 4- a1, b1 and c1 seem very similar, whereas a2, b2 and c2 differ significantly, can the authors explain why? Is every point in the scatter plots an MM structure? At lines 203-205 the authors state that two processes happen at the same time based on these plots which do not have time as a variable. Further explanation is needed here.

14. At line 235 "we also verified this idea", not sure which idea the authors are referring to, since they are writing before about their own results. This paragraph would strongly benefit from a conclusive sentence stating the relevance of the ESP analysis. 

Author Response

Response to Reviewer 2 Comments:

Point 1: Figure 1 should have the structure of the compounds studied and maybe the QM/MM zones, it would be clearer to the reader if they didn't have to scroll to page 5 to the mechanism picture.

Response 1: Your suggestion really means a lot to us. We have moved original Figure 1. “TTcluster was applied to RMSD and cluster analysis of 30 ns MD simulations performed with CHARMM” to Supporting Information. Insteadlly, “(a) Structural diagram of secondary structure of esterase PestE and (b) The chemical structure of methomyl and a respresentation of QM region” was named as Figure 1. This would be clearer to the readers. This change is in line 135. In addition, we have uploaded more quality image for all figures in the revised manuscript.

Point 2: In the computational details, the docking parameters used for Autodock should be specified, as well as the thermostat type for the MD simulations. In general, more details wouldn’t hurt in this section. 

Response 2: Thank you for the valuable comment. According to your suggestion, we have added the docking parameters used for Autodock in the computational details. “In the docking simulation, the grid box size was set 25 × 25 × 25 Å centered around the Ser 157 of esterase PestE. Nine docking models were obtained, and the best binding conformation was selected for the subsequent MD simulations.” This change is in line 88-91. Additionly, we have noted the thermostat type for the MD simulations. “a 30 ns stochastic boundary molecular dynamic (SBMD) simulation using canonical ensemble NVT at 298.15 K. “ This is in line 100.

Point 3: Figure 1 could be moved to the Supp Info or its results compressed to one single picture. 

Response 3: Thank for your nice suggestion. We have moved original Figure 1. “TTcluster was applied to RMSD and cluster analysis of 30 ns MD simulations performed with CHARMM” to Supporting Information.

Point 4: line 119, "more scientifically" should be changed by "thoroughly".

Response 4: Thank you the valuable suggestion. We have modified this sentence as suggested. This change is in line 131.

Point 5: I do not understand why the authors benchmark their energy error against a machine learning method. Which error does the ML method have? could the authors show error bars in their own results?

Response 5: Thank you for the valuable comment. Due to the advent of the era of big data, the machine learning method attracts a lot of attention. We compare the energy barrier using the existing machine learning data model with the Boltzmann-weighted average method, validating the viability of machine learning method. Because of the limitations of a single study, the error bars of machine learning method cannot be judged. In the future, we will conduct a wide range of studies to investigate the error bars of the existing machine learning data model and optimize the machine learning method. To sum up, the machine learning method used in this study will give important reference to the future research.

Point 6: Line 159 in "spacial structures" , spacial is redundant and should be removed.

Response 6: Thank for your careful checks. The “spacial” have been removed. This change is in line 169.

Point 7: Line 184, in "some chemical reactions", "some" sounds colloquial. The authors may consider rewriting that sentence.

Response 7: Thank you very much for the suggestion. We have changed “some” to “several”. The new modified sentence is “Noncovalent interactions dominate chemical interactions between a protein and a drug, or a catalyst and its substrate, and even several chemical reactions”. This change is in line 194.

Point 8: Line 187, the acronym RDG needs to be explained and after "electron density" the greek letter rho could be added to refer to the next equation.

Response 8: Thank you for your reminder. The acronym RDG has been explained: Reduced density gradient (RDG). This change is in line 197. Additionly, the greek letter rho has been added after “electron density” as suggested. This change is in line 198.

Point 9: Line 191, what is sign (lambda) ? In general the last three points show that the part of "Covalent Interactions" could be improved to make it more understandable to the readers.

Response 9: Thank you for the valuable comment. The lambda represents the second eigenvalue of the electron density Hessian matrix, and sign () represents the sign (Positive or negative value). This has been explained in the manuscript. Further, we have modified the part of the "Covalent Interactions" to make it more understandable to the readers.

Point 10: Line 209 refers to an unexistent Fig.4(b3).

Response 10: We were really sorry our careless mistakes. Thank you for your reminder. We have corrected “Fig.4(b3)” to “Fig.4(c2)”. This change is in line 221.

Point 11: Line 218: specied -> species.

Response 11: Thank you for your careful checks. We have changed “specied” to “species”. This change is in line 231.

Point 12: In Figure 4, the z axis seems to give the same information as the x axis. If this is not the case a clarification is needed.

Response 12: Thank you for the valuable comment. The z axis give the same information as the x axis. However, the color scale bars of z axis give more precise scale, which make readers easier to distinguish the color junction and know the precise value.

Point 13: Figures 4- a1, b1 and c1 seem very similar, whereas a2, b2 and c2 differ significantly, can the authors explain why? Is every point in the scatter plots an MM structure? At lines 203-205 the authors state that two processes happen at the same time based on these plots which do not have time as a variable. Further explanation is needed here.

Response 13: Thank you for the valuable comment.

Firstly, we will explain the difference of Figure 4(a1), (b1) and (c1). As shown in Figure 4(a1), the peak value ranges from -0.04 to -0.03 a.u., indicating that the attraction between O1 and C1, which promotes the occurrence of the nucleophilic attack reaction. However, the peak value shown in Figure 4(b1) and (b3) range from -0.02 to -0.01 a.u., which highlight the nucleophilic attack reaction shown in Figure(a1). This change is in line 209-211.

Secondly, every point in the scatter plots is not an MM structure. The principle of scatter plot is  dividing the QM region into thousands of grids, and noncovalent interaction of each grid corresponds to a point. We can combine the iso-surface graphs to know the areas of noncovalent interaction in the scatter plots.

Thirdly, we will explain two processes happen at the same time based on these plots which do not have time as a variable. From Fig. 4(b2)——Color-mapped reduced density gradient (RDG) iso-surface graphs of TS1, we can find that the interactions of proton (H1) transfer and nucleophilic attack reactions both exhibit a large red-blue annular region, which indicate that formation and breakage of chemical bonds of both regions. We revealed two processes happen at the same time from the atomic level.

Point 14: At line 235 "we also verified this idea", not sure which idea the authors are referring to, since they are writing before about their own results. This paragraph would strongly benefit from a conclusive sentence stating the relevance of the ESP analysis. 

Response 14: Thank you for the valuable comment. “we also verified this idea” refers to we have verified O3 will undergo a proton transferring process (step II) through QM/MM calculations. Electrostatic potential (ESP) is a concept in wave function analysis, which can predict reaction sites and molecular properties. And the intention of this sentence is that we have predicted the reaction site by ESP analysis. Further, we have verified this idea through QM/MM calculations. We have modified this sentence in the revised manuscipt. This change is in line 248-249.

Point: I am also missing a benchmark of several functionals or at least some solid justification to use the combination M06-2X/6-31G** (or the triple Z equivalent) for the QM/MM part.

Response: Thank you for the valuable comment. Our previous study has verfied that M06-2X/6-31G** for the QM/MM part is feasible to hydrolysis process (Chen et al., 2019; Feng et al., 2020).

Chen J, Wang J, Li Y, Wang X, Zhuang T, Zhang Q, et al. Catalysis mechanism of oxidized polyvinyl alcohol by pseudomonas hydrolase: Insights from molecular dynamics and QM/MM analysis. Chemical Physics Letters 2019; 721: 49-56.

Feng S, Yue Y, Chen J, Zhou J, Li Y, Zhang Q. Biodegradation mechanism of polycaprolactone by a novel esterase MGS0156: a QM/MM approach. ENVIRONMENTAL SCIENCE-PROCESSES & IMPACTS 2020; 22.

Round 2

Reviewer 2 Report

The manuscript is suitable for publication in the current form.